# Comparative Detection of Immunoglobulin Isotypes and Subclasses against *Toxoplasma gondii* Soluble Antigen in Serum and Colostrum Samples from Puerperal Women

**DOI:** 10.3390/ijerph19137953

**Published:** 2022-06-29

**Authors:** Hellen Dayane Silva Borges, Ana Carolina Morais Oliveira-Scussel, Ângela Maria Morais Oliveira, Vânia Olivetti Steffen Abdallah, Ana Cláudia Arantes Marquez Pajuaba, José Roberto Mineo

**Affiliations:** 1Laboratory of Immunoparasitology “Dr. Mário Endsfeldz Camargo”, Institute of Biomedical Sciences, Universidade Federal de Uberlândia, Uberlândia 38405-317, MG, Brazil; hellen.days@hotmail.com (H.D.S.B.); ana.morais@uftm.edu.br (A.C.M.O.-S.); ana.pajuaba@ufu.br (A.C.A.M.P.); 2Biomedicine Teaching Laboratory, Institute of Health Sciences, Federal University of Triângulo Mineiro, Uberaba 38025-180, MG, Brazil; 3Human Milk Bank, The Clinics Hospital of Universidade Federal de Uberlândia, Uberlândia, Minas Gerais, Uberlândia 38405-320, MG, Brazil; anacarolm@yahoo.com.br; 4Department of Pediatrics, The Clinics Hospital, Faculty of Medicine, Universidade Federal de Uberlândia, Uberlândia 38405-320, MG, Brazil; vosabdallah@hotmail.com

**Keywords:** *Toxoplasma gondii*, colostrum, serum, antibody isotypes

## Abstract

Background: *Toxoplasma gondii* is an obligate intracellular parasite that can infect several species, including humans, and can cause severe damage to the fetus when the infection occurs during pregnancy. The environment and/or food contamination are critical to spreading the infection. Human milk is rich in nutrients and bioactive elements that provide growth and development of the immune system of the newborn. All isotypes of immunoglobulins are present in human colostrum and they are produced from systemic or local sources. Breastfeeding protects the infant against various pathogens, but there is no conclusive study to detect IgG subclasses in colostrum against *T. gondii*. Therefore, the aim of this study was to detect and evaluate the presence of antibody isotypes against *T. gondii* in paired samples of serum and colostrum. Methods: The study included 283 puerperal patients. ELISA (Enzyme-Linked Immunosorbent Assay) for detection of anti-*T. gondii*-specific IgM, IgA, and IgG isotypes and IgG1, IgG3, and IgG4 subclasses were conducted on paired samples of serum and colostrum. Results: It was found that 45.9%, 6.0%, and 2.1% of serum samples and 45.2%, 7.1%, and 2.1% of colostrum samples were positive for IgG, IgM, and IgA, respectively. Specific IgG1, IgG3, and IgG4 were positive, respectively, in 98.5%, 54.6%, and 44.6% of serum samples, in contrast with 56.9%, 78.5%, and 34.6% of colostrum samples. Thus, the predominant reactivity of IgG subclasses against *T. gondii* was IgG1 in serum and IgG3 in colostrum. The higher percentage of positive samples and higher levels of anti-*T. gondii* IgG3 antibodies were observed in colostrum, when compared to serum samples, suggesting a local production of this subclass. IgG3 and IgG1 subclasses presented different percentages of positivity in serum and colostrum. Only the IgG1 subclass showed a significant correlation between the levels of anti-*T. gondii* in serum and colostrum, suggesting that IgG1 in breast milk comes from a systemic source. IgG4 showed a similar percentage of positivity in both sample types, but no significant correlation was observed between their levels. Conclusion: Colostrum presents representative levels of IgM, IgA, IgG1, IgG3, and IgG4 antibodies specific to *T. gondii*. The detection of these antibodies presents the potential for diagnostic application of colostrum samples to better identify the diagnostic status of *T. gondii* infection, especially during the acute phase. In addition, breastfeeding can also be a possible source of protective antibodies for the newborn against toxoplasmosis, an anthropozoonosis maintained by environmental infection, which interferes in the public health of many countries.

## 1. Introduction

*Toxoplasma gondii* is an obligate intracellular parasite that can infect several species, including humans, worldwide. Seroprevalence for *T. gondii* in the population varies according to the studied location. Previous studies have shown that about 40–80% of the Brazilian population is infected by *T. gondii* [1,2]. In Brazil, studies previously conducted reported a high prevalence of seropositive pregnant women for anti-*T. gondii* IgG in some states, highlighting 58.5% in Rio de Janeiro [3], 74.5% in Rio Grande do Sul [4], 69.3% in Sergipe [5], and 62% in São Paulo [6]. In Uberlândia, Minas Gerais state, a study showed that 51.6% of pregnant women showed seropositivity for *T. gondii* [7]. The environment and/or food contamination is critical to disseminate *T. gondii* infection [8,9]. Additionally, water is considered nowadays an additional important source of contamination [10,11]. 

During the infection in human beings, the production of immunoglobulins of IgM, IgG, IgA, IgD, and IgE isotypes has been reported, which can be used as tools for the diagnosis of toxoplasmosis. Concerning the kinetics of antibody production after primary infection, IgM and IgA are important isotypes to characterize the acute phase of infection. Even though the IgM isotype can be detected after 7–8 days of infection, it can persist for months in the serum samples of many patients. In contrast, IgA can be detected in the first weeks of infection, but its levels become undetectable after one or two months of infection. Usually, samples with positivity only for IgG antibodies are related to the chronic phase. The acute phase of infection is serologically characterized by the concomitant presence of IgG, IgM, and IgA, or the presence of IgG and IgM, or the presence of IgM and IgA in the absence of IgG antibodies. Additionally, IgG antibodies represent the major immunoglobulin isotype involved in the anti-*T. gondii* humoral response. In human serum samples, *T. gondii* antigen-specific IgG antibodies appear after 1 or 2 weeks of infection and their levels increase to a peak within 6–14 months after infection [12]. 

*T. gondii* antigen-associated antibodies can be detected in a variety of biological fluids, such as blood serum, cerebrospinal fluid, or human milk [13,14]. Human milk contains more than 200 different substances, a homogeneous mixture consisting of lipids, proteins, carbohydrates, vitamins, minerals, various cell types, and water whose composition varies according to the stage of lactation. There are three different stages: colostrum, transitional milk, and mature milk [15,16]. Colostrum is a yellowish-colored secretion due to its high beta-carotene content, it is particularly rich in immunoglobulins, antimicrobial peptides, and other bioactive molecules, including immunomodulatory and anti-inflammatory substances [17]. Colostrum is considered to be milk produced from the 1st to the 7th day after childbirth [18]. Transitional milk is secreted between the 7th and 14th day of lactation and presents some of the characteristics of colostrum. This stage represents a ramp-up period to fully mature milk production. Mature milk begins to be produced on the 15th day after childbirth, and it is rich in all the nutrients needed to feed and support the needs of rapid growth and child development [19,20].

It has been demonstrated that human milk provides highly effective protection against a variety of local or systemic infections [21,22] and contains numerous cytokines and immunomodulatory factors, some of them at higher concentrations than those found in circulating blood. There are also factors such as cytokines, which are mediators of non-specific immunity, that have a stimulatory effect on T and B lymphocytes and stimulate IgA production, but also have inhibitory and modulating effects [15,23]. Given this, in recent years several studies have focused on the identification of immune substances present in human milk that act as a protective factor [24,25,26].

All immunoglobulin isotypes are present in human milk, especially colostrum and can be produced from systemic or local sources. However, to date there are no conclusive studies to detect *T. gondii*-directed IgG subclasses in colostrum. Therefore, due to the high prevalence of this disease as well as the importance of breastfeeding, the aim of this study was to evaluate the presence of *T. gondii*-specific IgG, IgM, and IgA as well as IgG subclasses 1, 2, and 4 in paired serum and colostrum samples.

## 2. Methods

### 2.1. Patients

The present study involved 283 postpartum women, who were hospitalized due to childbirth at the Obstetric Center of The Hospital de Clínicas da Universidade Federal de Uberlândia and accepted to participate in the study after being duly clarified. The exclusion criteria of the study consisted of HIV-positive and/or human cell lymphotropic virus (HTLV) seropositive women, for whom breastfeeding is contraindicated by the Ministry of Health [27]. The laboratory study involving the serological analysis of the mothers was carried out at the Immunoparasitology Laboratory of the Biomedical Sciences Institute of the Federal University of Uberlândia. The study followed the declaration of Helsinki, with its protocol approved by Ethical Committee of Research da Universidade Federal de Uberlândia (Protocol # 104/110). Written consent was obtained from each patient.

### 2.2. Human Colostrum Samples

Samples of human colostrum were collected by manual milking at an approximate volume of 3 mL of each puerperal woman. The samples were collected from postpartum women at the Obstetric Center of the Uberlândia Federal University Hospital, between the 1st and 6th day after childbirth, to ensure that the sample was actually human colostrum. Lipid removal was performed by processing the colostrum samples by centrifugation (500× *g*/10 min at 4 °C), the skim milk was aliquoted and stored at −20 °C until use.

### 2.3. Blood Samples

The Clinical Analysis Laboratory of Uberlândia Federal University Hospital routinely collects blood from all hospitalized women one day after delivery for laboratory tests. Therefore, the remaining blood volume of the puerperal women who signed the “Free and Informed Consent Form” was requested from the laboratory. Thus, no further blood collection was required from the mothers who agreed to participate in the study. The obtained serum was aliquoted and stored at −20 °C until the serological tests were performed.

### 2.4. Maintenance of T. gondii

The tachyzoite forms of the *T. gondii* RH strain were maintained in cell culture, as well as purified to remove debris from host cells, using HeLa cell lines and the protocol described previously [28]. Briefly, cells infected with *T. gondii* tachyzoites were maintained by serial passages in RPMI (Roswell Park Memorial Institute) medium with 2% bovine fetal serum, every 48–72 h. Free parasites were collected by removing the cell monolayer from culture flasks and partially purified by forced passage through a 13 × 4 mm needle and rapid centrifugation (45× *g* for 1 min at 4 °C) to remove cell debris. The supernatant was collected and washed twice (700× *g* for 10 min at 4 °C) with 0.01 M phosphate buffered saline (PBS, pH 7.2). The final pellet of the parasite suspension was resuspended in PBS and the parasites counted in Neubauer’s chamber using 0.4% Trypan blue vital exclusion dye (Sigma Chemical Co., St. Louis, MI, USA). The parasites were stored at −20 °C until preparation of the soluble *T. gondii* antigen (STAg).

### 2.5. Indirect ELISA for Detection of Anti-T. gondii IgG Total and Subclasses

The presence of IgG antibodies and their subclasses (IgG1, IgG3, and IgG4) to *T. gondii* were investigated in paired serum and colostrum samples from puerperal women using the iELISA, performed as previously described, with modifications [29]. The detection of IgG1, IgG3, and IgG4 subclasses in serum and colostrum samples were performed only in samples from puerperal women who positively tested for anti-*T. gondii* IgG. High affinity microplates (Corning Incorporated Costar^®^, Corning Laboratories Inc., New York, NY, USA) were sensitized with STAg (10 μg/mL) diluted in 0.06 M carbonate-bicarbonate buffer (pH 9.6) and incubated for 18 h at 4 °C. Subsequently, the plates were washed with PBS containing 0.05% Tween 20 (PBS-T) and blocked with PBS-T containing 5% skimmed-milk powder (Molico, Nestlé, São Paulo, SP-PBS-T-M 5%), for 1 h at room temperature. After washing, serum samples were diluted 1:64 in PBS-T-M 5%, and colostrum samples were diluted 1:5 in PBS-T and incubated for 1 h at 37 °C. After six wash cycles, secondary antibodies were added, which consisted of the following conjugates: anti-human IgG labelled with peroxidase (1:2000); biotinylated anti-IgG1, anti-IgG3 and anti-IgG4 (Sigma Chemical Co., St. Louis, MI, USA), diluted 1: 2000, 1:2000, and 1:4000, respectively. Amplification of the reaction signal was performed by the addition of streptavidin-peroxidase (Sigma Chemical Co., St. Louis, MI, USA). The reaction was revealed by the addition of the Peroxidase Substrate System (ABTS, Kirkegaard and Perry Laboratories, KPL, Washington, DC, USA). Optical density (OD) values were determined in a microtiter plate reader (Titertek Multiskan Plus MKII, Flow Laboratories, McLean, VA, USA) at 405 nm. Positive and negative controls were included on the plate. The levels of antibodies were arbitrarily expressed in an ELISA index (IE), according to the formula: IE = OD sample/cut off, where cut off was calculated as the mean OD of negative control sera plus three standard deviations. IE ≥ 1.2 values were considered positive to exclude borderline reactivity values close to IE values = 1.0.

### 2.6. Capture ELISA for Detection of Anti-T. gondii IgM and IgA

Capture Enzyme Linked Immunosorbent Assay (cELISA) for the detection of IgM and IgA antibodies present in paired serum samples and colostrum samples was performed as previously described [29], with modifications. Briefly, high affinity microplates (Corning Incorporated Costar^®^, Corning Laboratories Inc., New York, NY, USA) were sensitized with anti-human IgM capture antibodies (1:200) (Kirkegaard and Perry Laboratories, KPL, Washington, DC, USA) or anti-human IgA (1:100) (Sigma Chemical Co., St. Louis, MI, USA) in 0.06 M carbonate-bicarbonate buffer (pH 9.6) and incubated for 18 h at 4 °C. The plates were blocked with PBS-T-M 5% for 1 h at room temperature. Subsequently, serum samples were diluted 1:16 in PBS-T-M 5%, while colostrum samples were diluted 1: 5 in PBS-T and incubated for 2 h at 37 °C. The plates were incubated with STAg (100 μg/mL) in PBS-T-M 5% for 2 h at 37 °C. Bound antigen was detected by the addition of rabbit anti-*T. gondii* antibody conjugated with peroxidase [30] diluted 1:500 in PBS-T-M 5%, followed by incubation for 1 h at 37 °C. The final steps were as described for indirect ELISA. Between each stage of the reaction, washing cycles were performed with PBS-T.

### 2.7. Statistical Analysis

To better characterize serological results based on the reactivity of IgG1, IgG3, and IgG4 specific against *T. gondii*, samples were divided into three different groups: **Group 1**: serum and colostrum samples from puerperal women with acute *T. gondii* infection, characterized by positive results for specific IgG, IgM, and IgA antibodies (IgG+/IgM+/IgA+); **Group 2**: serum and colostrum samples from puerperal women with positive results for IgG and IgM antibodies specific for *T. gondii* (IgG+/IgM+/IgA−); **Group 3**: serum and colostrum samples from puerperal women with chronic *T. gondii* infection, characterized by only positive results for specific IgG (IgG+/IgM−/IgA−). To better define the diagnostic status, IgG1 and IgG3 antibodies were evaluated, characterizing two possible outcomes: early acute phase and convalescent phase, as previously described. The early acute phase (0–3 months of infection) has an IgG3/IgG1 ratio >1.0 with positive IgM and/or IgA antibodies. The convalescent phase (3–12 months of infection) has an IgG3/IgG1 ratio <1.0 with positive IgM and/or IgA.

Contingency analyses were performed to evaluate the differences between positive samples, using the Fisher’s exact test. Differences between levels of the same antibody isotypes in human serum and colostrum samples were analyzed using Wilcoxon test. The differences between the median ELISA index levels of the subclasses of IgG1, IgG3, and IgG4 in non-normal distribution sampling were analyzed using the Friedman test and Dunn’s multiple comparison post-test for paired samples. Correlation analyses were performed using the Spearman tests, according to the Gauss distribution of the sample. Values of *p* < 0.05 were considered statistically significant.

## 3. Results

### 3.1. Detection of T. gondii-Specific IgG, IgM, and IgA Antibodies in Paired Serum and Colostrum Samples

*T. gondii*-specific IgG, IgM, and IgA antibodies were evaluated by ELISA in 283 paired serum and colostrum samples. As shown in Figure 1, the percentages of positive results for IgG, IgM, and IgA, based on the ELISA indices (EI) were: 130 (45.9%), 17 (6.0%), and 6 (2.1%) for positive serum samples, respectively; and 128 (45.2%), 20 (7.1%), and 6 (2.1%) of positive colostrum samples for IgG, IgM, and IgA, respectively. The positivity (%) of IgG, IgM, and IgA for serum samples were similar to the colostrum samples.

The levels of IgG antibodies to *T. gondii* detected in colostrum samples were significantly higher than IgG detected in serum samples (*p* < 0.0001). In contrast, the levels of IgM and IgA antibodies detected in colostrum samples were significantly lower than IgM and IgA detected in serum samples (*p* < 0.0001). No statistical difference among the levels of IgG, IgM, and IgA isotypes to *T. gondii* was found in serum samples, while IgG levels were significantly higher than IgM to IgA in colostrum (*p* < 0.0001).

### 3.2. Detection of T. gondii-Specific IgG1, IgG3, and IgG4 Subclasses in Paired Serum and Colostrum Samples

Subclasses IgG1, IgG3, and IgG4 specific to *T. gondii* were detected by indirect ELISA in paired serum and colostrum from all puerperal women for anti-*T. gondii* IgG positives (*n* = 130), as shown in Figure 2.

Positivity rates (%) for the IgG1 subclass detected in serum samples (*n* = 128, 98.5%) were significantly higher than for IgG1 detected in colostrum (*n* = 74; 56.9%) (*p* < 0.0001). In contrast, IgG3 antibodies detected in colostrum (*n* = 102; 78.5%) showed significantly higher levels than in serum samples (*n* = 71; 54.6%) (*p* = 0.0009). No difference was found for the IgG4 subclass between serum (*n* = 58; 44.6%) and colostrum (*n* = 45; 34.6%) samples (*p* = 0.1938).

ELISA index to evaluate IgG1 levels in serum samples (median = 4.23) presented statistically higher values, when compared to IgG3 and IgG4 levels (medians = 1.20 and 1.13, respectively) (*p* < 0.0001). No difference between IgG3 and IgG4 antibody levels was found in serum. In colostrum samples, however, IgG3 presented higher levels (median = 1.93) than IgG1 (median = 1.28) (*p* < 0.05) and IgG4 (median = 1.05) (*p* < 0.001). Concerning the comparison between IgG1 and IgG4 levels in colostrum, a significant difference was also found (*p* < 0.01). Moreover, the levels of ELISA index for IgG1 antibodies against *T. gondii* detected in serum samples were significantly higher when compared to paired samples of colostrum (*p* < 0.0001). In contrast, significantly higher levels of ELISA index for IgG3 antibodies to this parasite were found in colostrum samples (*p* < 0.0001), when compared to paired serum samples. There was no statistical difference between IgG4 levels in serum and colostrum samples (*p* = 0.9146).

### 3.3. Correlation between Anti-T. gondii IgG1, IgG3, and IgG4 Subclass Levels in Serum and Colostrum Samples

The correlation analysis between specific anti-*T. gondii* IgG subclasses in serum and colostrum samples showed a significant positive correlation when IgG1 levels were compared (*r* = 0.3409; *p* < 0.0001), (Figure 3A). No significant correlation was observed for IgG3 and IgG4 subclasses when analyzed in serum and colostrum samples (Figure 3B,C).

### 3.4. Associations between IgG1, IgG3, and IgG4 Subclasses and T. gondii-Specific IgG, IgM, and IgA Isotypes in Serum and Colostrum Samples

The association between serological results based on the reactivity of *T. gondii*-specific IgG1, IgG3, and IgG4 was evaluated regarding IgM and/or IgA positivity, as shown in Table 1. In total, 123 samples demonstrated simultaneous reactivity for anti-*T. gondii* IgG, IgM, or IgA in serum and colostrum. These samples were divided into three groups: Group I (*n* = 2) serum and colostrum samples from puerperal women with acute *T. gondii* infection, characterized by positive results for specific IgG, IgM, and IgA antibodies (IgG+/IgM+/IgA+); Group II (*n* = 10) serum and colostrum samples from puerperal women with positive results for IgG and IgM antibodies specific for *T. gondii* (IgG+/IgM+/IgA−). Group III (*n* = 111) serum and colostrum samples from postpartum women with chronic *T. gondii* infection, characterized by only a positive result for specific IgG (IgG+/IgM−/IgA−).

Group I was constituted by just two serum and colostrum samples from postpartum women positive for *T. gondii*-specific IgG, IgM, and IgA, being 100% (*n* = 2) of serum samples positive for IgG1, IgG3, and IgG4. In colostrum, all samples were positive for IgG3 and IgG4 and one of the colostrum samples was negative for IgG1.

Group II consisted of 10 paired serum and colostrum samples from postpartum women with positive results for *T. gondii*-specific IgG and IgM, presented 100% (*n* = 10), 50% (*n* = 5), and 40% (*n* = 4) of serum and 70% (*n* = 7), 90% (*n* = 9), and 40% (*n* = 4) of colostrum samples positive for IgG1, IgG3, and IgG4, respectively (Figure 4A). IgG1 and IgG3 presented a statistically significant difference in positivity (%) between serum and colostrum samples (*p* < 0.0001). Serum levels of IgG1 showed statistically higher levels than IgG3 (*p* < 0.05) and IgG4 (*p* < 0.001); however, there was no statistically significant difference in EI levels between subclasses in colostrum.

Group III was composed of 111 serum and colostrum samples from postpartum women with chronic *T. gondii* infection, characterized by the positive result for IgG specific. It was found that 98.2% (*n* = 109), 53.2% (*n* = 59), and 44.1% (*n* = 49) of serum samples and 54,1% (*n* = 60), 76.6% (*n* = 85), and 35.1% (*n* = 39) of colostrum samples were positive for IgG1, IgG3, and IgG4, respectively (Figure 4B). Comparing the positivity rate (%) for each of the subclasses in serum and colostrum samples, a statistical difference was observed only for IgG1 (*p* < 0.0001) and IgG3 (*p* = 0.0006). When analyzing the differences in EI levels between the subclasses, it was observed that IgG1 levels were statistically higher when compared to serum IgG3 (*p* < 0.0001) and IgG4 (*p* < 0.0001). In colostrum, IgG3 levels were statistically higher than IgG1 (*p* < 0.05) and IgG4 (*p* < 0.01). There was also a significant difference between IgG1 and IgG4 (*p* < 0.0001) in colostrum.

As shown in Table 2, the results obtained after the analysis of 130 serum and colostrum samples from puerperal women, based on the simultaneous detection of IgM, IgA, IgG isotypes, as well as IgG subclasses, particularly with the inclusion of the IgG3/IgG1 ratio, help assemble the diagnosis status into three different phases, as follows: chronic, early acute, and convalescent phase. The information presented in Table 3 summarizes the results from 16 cases of puerperal women and their association with the suggested diagnosis status, when serum and colostrum samples were analyzed simultaneously.

## 4. Discussion

Antibodies from the IgG isotype in healthy adults represent approximately 75% of total serum immunoglobulins. Four subgroups of the human IgG heavy chain were first identified as antigenically distinct by using polyclonal antisera prepared against human IgG myeloma proteins [31]. These subtypes were later classified as IgG subclasses 1, 2, 3, and 4 depending on their concentration in normal serum and the frequency of their occurrence from the myeloma proteins [32]. The subclasses differ in structure, genetic characteristics, susceptibility to enzymatic digestion, and catabolism [33,34]. They have different profiles of effector functions, in particular the ability to mediate complement activation and bind to receptors on the surface of cells involved in the immune response [35,36,37].

In the present study, we evaluated the presence of IgG, IgM, and IgA isotypes, as well as IgG1, IgG3, and IgG4 subclasses in response to infection by *T. gondii* in paired samples of serum and colostrum from postpartum women. It was observed that 98.5% of serum samples showed reactivity to IgG1, indicating the predominance of IgG1 activity in serum against this infection. These data are in agreement with reports by Huskinson et al. (1989) and Cañedo-Solares et al. (2008), who stated that in humans the predominant IgG antibody response to infection with *T. gondii* is from the IgG1 subclass [38,39]. Another study demonstrated that IgG1 antibodies present in serum have high sensitivity (71%) and specificity (91.4%), suggesting its potential use as a biomarker of ocular toxoplasmosis [40]. Even though previous studies have already shown that all four IgG subclasses can cross the placenta to protect the fetus from infections, there is a higher rate of IgG1 vertical transport when compared to other IgG subclasses. In colostrum, however, only around half of the samples (56.9%) were positive for IgG1. It can be hypothesized that the lower presence of this immunoglobulin subclass in maternal colostrum antibodies specific to *T. gondii*, could be counterweighed by the high maternal–neonatal transfer of anti-*T. gondii* IgG1 through the placenta.

When analyzing the detection of IgG subclasses in colostrum in comparison with the total IgG content, it has already been described that IgG1 was predominant, followed by IgG2, IgG4, and IgG3 [41,42,43]. Interestingly, the present work showed a predominance of IgG3 specific to *T. gondii* in colostrum in relation to the other subclasses, 78.5% of the samples being positive. Specific-*T. gondii* IgG3 has also been detected at higher levels in saliva in samples paired with serum, suggesting local production by cells of the salivary glands [44]. Similarly, the data from the present report reinforce the hypothesis that the presence of detectable levels of IgG3 in colostrum may be the result of a selective secretion and/or due to a local production of this subclass by the mammary gland cells. Analyzing serum samples, it was observed that more than half of the mothers (54.6%) were reactive to anti-*T. gondii* IgG3, However, Cañedo-Solares et al. (2008), in their work, were unable to detect with a higher percentage of accuracy the cases of mothers infected by *T. gondii* based also on the seropositivity for IgG3 [39].

The kinetic profile of anti-STAg IgG1 and IgG3 in individuals infected with *T. gondii* has been described in the literature and it has been observed that the levels of these antibodies increase over time: IgG1 levels increase until the 6th month of infection, followed by a slight decrease and maintenance of levels up to the 12th month of infection, whereas IgG3 levels increase only from the 3rd to the 6th month of infection [45]. Additionally, De Souza-e-Silva (2013) studied the association between clinical signs of congenital toxoplasmosis and IgG subclasses found in newborns and observed that anti-*T. gondii* IgG3 was associated with newborns without neurologic damage, highlighting the importance of this immunoglobulin in protecting the newborn from infection by *T. gondii* [46].

Our study demonstrated that the group of seropositive mothers for IgG and IgM and the group of chronically infected women with *T. gondii* also showed predominant reactivity to IgG1 in serum and IgG3 in colostrum. These results are in accordance with the literature, which reports that there is a predominance of IgG1 and IgG3 subclasses in the antibody response against the immunodominant parasite antigens [34]. IgG1 and IgG3 subclasses play an important role during infection, being highly related to the parasite lysis because they are able to activate the complement system, bind to IFN-γ activating NK cells, macrophages and neutrophils, and have opsonizing activity [47].

The present study included the results of 16 cases of puerperal women with diagnostic status of acute toxoplasmosis. Paired samples of serum or colostrum from these patients were analyzed to characterize the diagnostic status of *T. gondii* infection. When assessing these results, it is clear that new pieces of information were obtained. In fact, the diagnostic status based on the results from colostrum samples of nine cases indicated a profile of more recent infection, when compared to those obtained from serum samples; five showed a similar diagnostic status for both colostrum and serum samples; and two had a profile indicating a later diagnostic status of *T. gondii* infection when colostrum results were compared to serum. Taken together, these results support the idea of using colostrum samples as a useful diagnostic tool to determine the status of *T. gondii* infection, especially during the acute phase.

The total serum concentration of IgG4 in adults corresponds to 4% of total IgG [48]. Despite the low proportion of total IgG4, our results suggest that IgG4 antibodies play an important role in the response against *T. gondii* infection, since we found 44.6% and 34.6% positivity in serum and colostrum samples, respectively. Our results differ from what was presented by Huskinson (1989), who was unable to detect anti-*T. gondii* IgG4 through immunoenzymatic assays [34]. Our data support a study with recombinant proteins as a diagnostic target for toxoplasmosis that demonstrated considerable levels of IgG4 reactive with rMIC3 [49].

It has been suggested that antibodies of the IgG4 subclass are important in the total IgG response, especially when exposure to the antigen is chronic [50,51]. There are reports, however, pointing out that anti-*T. gondii* IgG4 antibodies were not detected in sera from individuals with latent infection. Although IgG4 antibodies reflect prolonged antigenic stimulation, the presence of this immunoglobulin subclass does not suggest a beneficial response profile [52]. IgG4 antibodies interfere in the protective immunity of infected individuals, mainly by IFN-γ inhibition, but also because complement is not activated, which reduce the effector function of this subclass compared to other IgG subclasses, as its production is driven by Th2 cytokines, such as IL-4, IL-13, and IL-10 [53,54]. A study conducted with individuals with the HIV virus showed that IgG4 levels against *T. gondii* are significantly higher in patients with cerebral toxoplasmosis, suggesting that IgG4 may be valuable to support the diagnosis of focal brain lesions caused by *T. gondii* infection, in seropositive patients [55]. This approach can be useful, especially when molecular research to detect parasites is not available.

In the present study, the percentage of anti-*T. gondii* IgG4 positivity in serum was similar to colostrum; however, there was no correlation between serum ELISA index and the levels observed in colostrum. For anti-*T. gondii* IgG1 and IgG3 the percentage of positivity distribution among the subclasses in colostrum was different from that presented in serum of the mothers. Similar results were found for subclasses of total IgG in studies with serum and colostrum [56,57]. The work developed by Guglietta et al. (2007) showed that newborns diagnosed with congenital toxoplasmosis generally have a helper T cell hyperresponsiveness, which is important in protecting and controlling infection against the parasite. According to the authors, the acquisition of an effective immune response is age dependent, that is, it progressively increases during growth and it is only around the age of four when children acquire immunity similar to the response of immunocompetent adults with *T. gondii* infection [53,58,59,60]. Therefore, newborns need to receive maternal antibodies, through the placenta and breast milk, to protect against various infections, including *T. gondii* infection.

## 5. Conclusions

Considering that maternal IgG antibodies play a protective role in congenital toxoplasmosis, reducing maternal parasitemia in the placenta and can also be passively transferred to the fetus through the placenta, it can be concluded that the results obtained in the present study demonstrated that maternal colostrum has significant levels of IgG1, IgG3, and IgG4 subclasses specific to *T. gondii*, and that breastfeeding can also be a possible source of protective antibodies for the newborn against toxoplasmosis, an anthropozoonosis maintained by environmental infection, which interferes in the public health of many countries.

## Figures and Tables

**Figure 1 ijerph-19-07953-f001:**
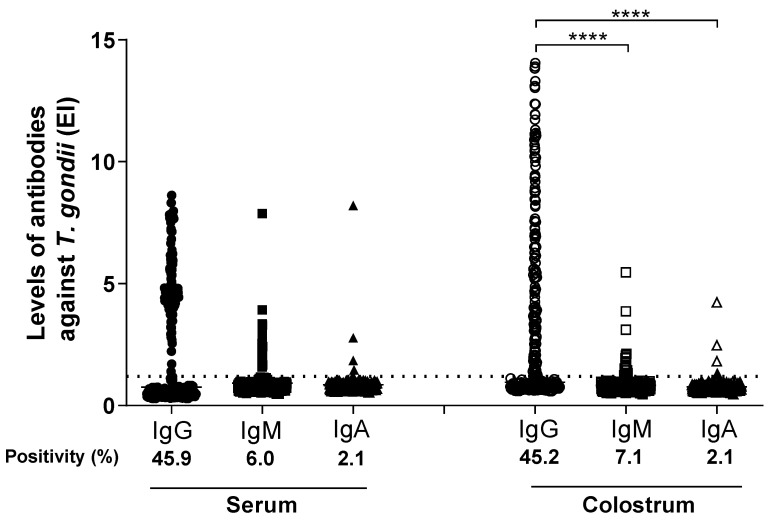
Levels of IgG, IgM, and IgA antibodies against *T. gondii*. Data determined by ELISA index (EI), in paired samples of serum and colostrum of postpartum women (*n* = 283). The solid lines represent the median values of the IE for each reaction and the dashed line the cut-off value (EI = 1.2). Differences between EI levels were calculated by Dunn’s multiple comparison test (**** *p* < 0.0001).

**Figure 2 ijerph-19-07953-f002:**
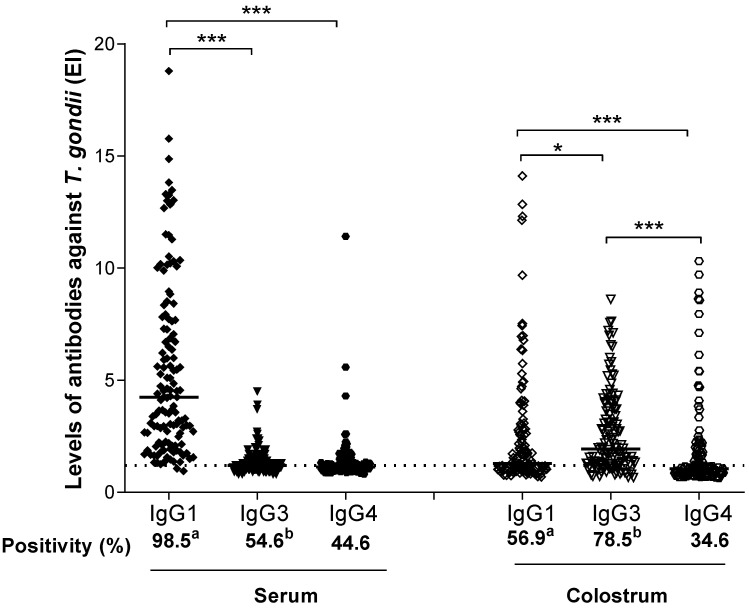
Levels of IgG1, IgG3, and IgG4 antibodies against *T. gondii*. Data determined by ELISA index (EI), in serum and colostrum samples from 130 puerperal women. The dashed line represents the cut-off value (EI = 1.2). The solid lines represent the values of the medians of the EI for each reaction. Differences between EI levels were calculated by Dunn’s multiple comparison test (* *p* < 0.05; *** *p* < 0.001). The differences between the levels of positivity (%) (a and b) of the samples were calculated using the Fisher’s exact test. Equal letters indicate statistical significance (*p* < 0.05).

**Figure 3 ijerph-19-07953-f003:**
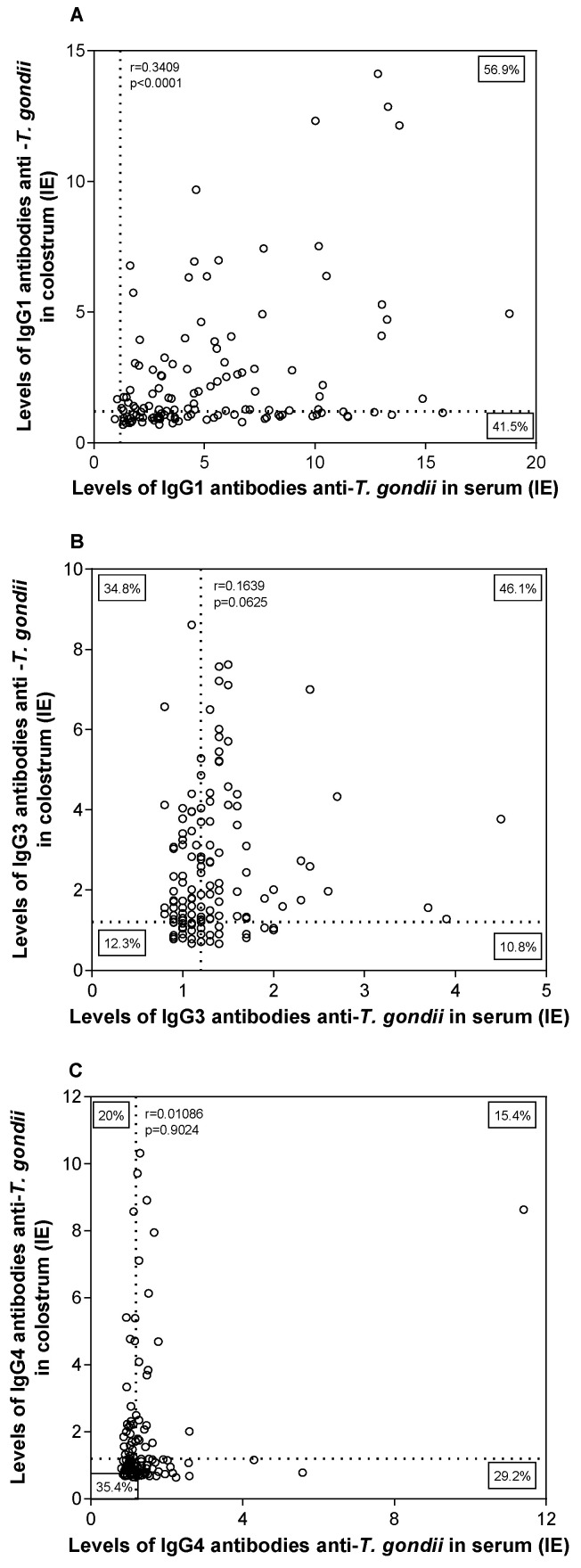
Correlation between levels of IgG1 (**A**), IgG3 (**B**), and IgG4 (**C**) antibody subclasses against *T. gondii*. Data determined by ELISA index (IE) present in paired serum and colostrum samples of 130 puerperal women. Dashed lines indicate the positivity cut off value (IE > 1.2). Double-positive, double-negative, or single-positive percentages are indicated in each corresponding corner. Correlation coefficients (*r*) were calculated by Spearman correlation test (*p* < 0.05).

**Figure 4 ijerph-19-07953-f004:**
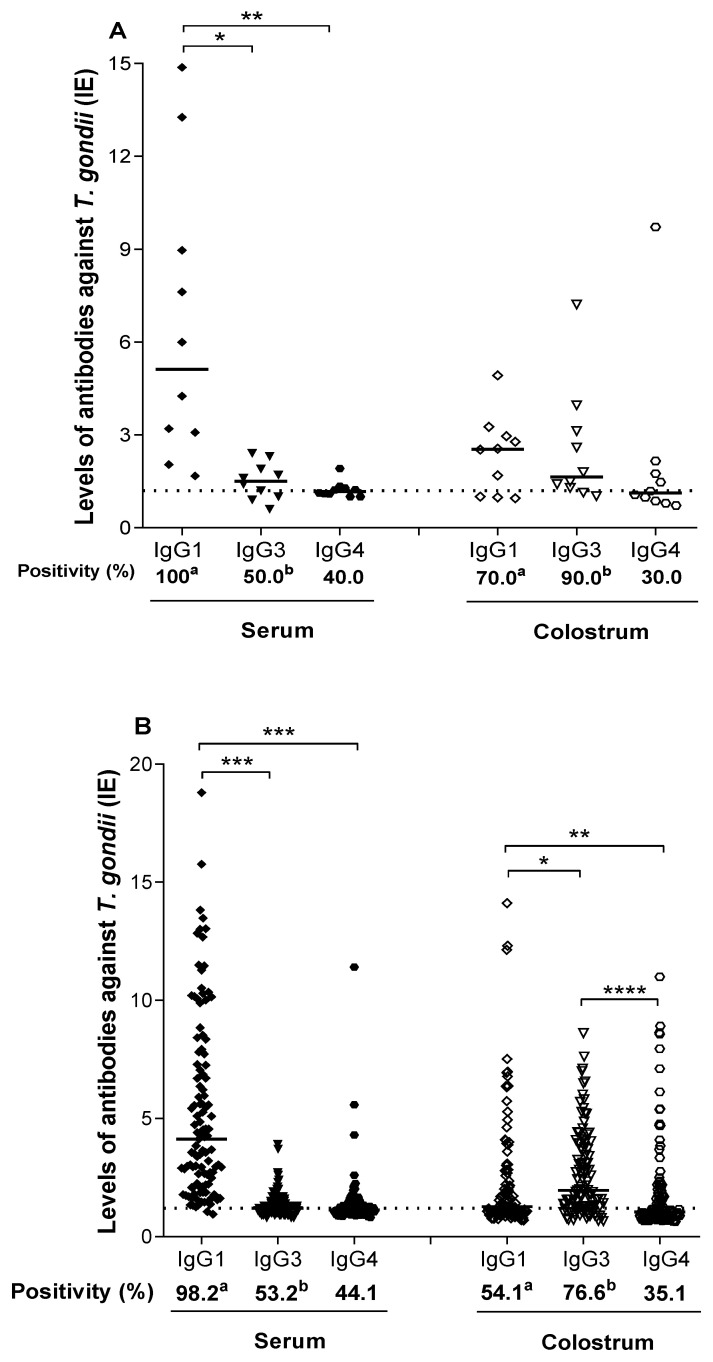
Levels of IgG1, IgG3, and IgG4 antibody subclasses against *T. gondii* in groups of paired samples. Data determined by ELISA index (IE), present in paired serum and colostrum samples. (**A**) Group II: levels of anti-*T. gondii* IgG1, IgG3, and IgG4 subclasses, of 10 samples from positive puerperal women for IgG and IgM antibodies specific to *T. gondii*. (**B**) Group III: levels of anti-*T. gondii* IgG1, IgG3, and IgG4 subclasses, in samples from 111 puerperal women with chronic toxoplasmosis. The dashed line represents the cut-off value (EI = 1.2). The solid lines represent the values of EI medians for each reaction. Differences between EI levels were calculated by Dunn’s multiple comparison test (* *p* < 0.05; ** *p* < 0.01; *** *p* < 0.001; **** *p* < 0.0001. The differences between the levels of positivity (%) (a and b) of the samples were calculated using Fisher’s exact test. Equal letters indicate significance (*p* < 0.05).

**Table 1 ijerph-19-07953-t001:** Association among the levels of IgG1, IgG3, and IgG4 subclasses specific to *T. gondii* and IgG, IgM, and IgA isotypes in serum and colostrum samples from 123 puerperal women from Uberlândia, MG, Brazil. Data are presented in number (*n*) and percentage (%) of the analyzed samples.

			Serum *n* (%)			Colostrum *n* (%)	
Association of Isotypes IgG/IgM/IgA		IgG1	IgG3	IgG4	IgG1	IgG3	IgG4
+/+/+	*n* = 2	2 (100)	2 (100)	2 (100)	1 (50.0)	2 (100)	2 (100)
+/+/−	*n* = 10	10 (100)	5 (50.0)	4 (40.0)	7 (70.0)	9 (90.0)	3 (30.0)
+/−/−	*n* = 111	109 (98.2)	59 (53.2)	50 (44.1)	60 (54.1)	85 (76.6)	39 (35.1)
**Total**		**121 (98.4)**	**66 (53.6)**	**56 (45.5)**	**68 (55.2)**	**96 (78.0)**	**44 (35.8)**

**Table 2 ijerph-19-07953-t002:** Diagnosis status from 130 serum and colostrum samples of puerperal women from Uberlândia, MG.

	Chronic Phase	Early Acute Phase 0–3 Months of Infection	Convalescent Phase 3–12 Months of Infection
	IgG+	IgG3/IgG1 > 1 IgM+IgA+ or IgM+IgA−	IgG3/IgG1 < 1 IgM+IgA+ or IgM+IgA−
**Serum** ***n* (%)**	117 (90.0%)	1 (0.77%)	12 (9.23%)
**Colostrum** ***n* (%)**	115 (88.46%)	6 (4.62%)	9 (6.92%)

Brazil. Data are presented in number (*n*) and percentage (%) of the analyzed samples.

**Table 3 ijerph-19-07953-t003:** Results of IgM and IgA levels in addition to IgG3/IgG1 ratios in serum and colostrum samples from 16 cases of puerperal women with acute status diagnosis in serum or colostrum samples.

	SERUM				COLOSTRUM			
Case	IgG3/IgG1 Ratio	IgM (EI)	IgA (EI)	Diagnosis Status	IgG3/IgG1 Ratio	IgM (EI)	IgA (EI)	Diagnosis Status
1	0.31	Neg (0.59)	Neg (0.55)	Chronic infection	0.14	Pos (1.26)	Neg (0.78)	Convalescent phase 3–12 months of infection
2	0.25	Pos (1.58)	Neg (0.67)	Convalescent phase 3–12 months of infection	0.43	Pos (1.32)	Neg (0.82)	Convalescent phase 3–12 months of infection
3	1.16	Pos (3.14)	Neg (0.69)	Early acute infection 0–3 months of infection	0.88	Pos (1.35)	Neg (0.99)	Convalescent phase 3–12 months of infection
4	0.24	Pos (3.36)	Neg (1.01)	Convalescent phase 3–12 months of infection	2.86	Pos (3.11)	Neg (0.81)	Early acute infection 0–3 months of infection
5	0.75	Pos (2.76)	Pos (1.85)	Convalescent phase 3–12 months of infection	7.07	Pos (3.86)	Pos (1.82)	Early acute infection 0–3 months of infection
6	0.21	Pos (1.93)	Neg (1.07)	Convalescent phase 3–12 months of infection	0.64	Pos (1.28)	Neg (0.71)	Convalescent phase 3–12 months of infection
7	0.34	Pos (2.29)	Pos (2.77)	Convalescent phase 3–12 months of infection	0.29	Pos (1.24)	Pos (1.30)	Convalescent phase 3–12 months of infection
8	0.55	Pos (1.62)	Neg (0.64)	Convalescent phase 3–12 months of infection	0.50	Pos (1.52)	Neg (0.83)	Convalescent phase 3–12 months of infection
9	0.15	Pos (3.30)	Neg (1.12)	Convalescent phase 3–12 months of infection	1.03	Pos (1.82)	Neg (0.82)	Early acute infection 0–3 months of infection
10	0.12	Neg (0.63)	Neg (0.86)	Chronic infection	0.32	Pos (1.94)	Neg (0.82)	Convalescent phase 3–12 months of infection
11	0.63	Pos (1.66)	Neg (0.88)	Convalescent phase 3–12 months of infection	1.23	Pos (1.29)	Neg (0.86)	Early acute infection 0–3 months of infection
12	0.12	Pos (2.18)	Neg (0.77)	Convalescent phase 3–12 months of infection	0.28	Pos (1.22)	Neg (0.86)	Convalescent phase 3–12 months of infection
13	0.27	Pos (2.28)	Neg (1.01)	Convalescent phase 3–12 months of infection	3.11	Pos (1.61)	Neg (0.90)	Early acute infection 0–3 months of infection
14	0.19	Neg (1.18)	Neg (0.88)	Chronic infection	0.43	Pos (1.35)	Neg (0.72)	Convalescent phase 3–12 months of infection
15	0.30	Pos (1.91)	Neg (0.87)	Convalescent phase 3–12 months of infection	4.02	Pos (1.25)	Neg (0.58)	Early acute infection 0–3 months of infection
16	0.12	Pos (2.15)	Neg (1.08)	Convalescent phase 3–12 months of infection	0.29	Neg (0.97)	Neg (0.91)	Chronic infection

## Data Availability

All data regarding this study can be obtained under request to corresponding author.

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
