# Peer review of "Comparative Detection of Immunoglobulin Isotypes and Subclasses against Toxoplasma gondii Soluble Antigen in Serum and Colostrum Samples from Puerperal Women"

_ijerph, 2022, doi:10.3390/ijerph19137953_

Round 1

Reviewer 1 Report

This is an interesting article, it would be worth publishing, however, linguistic inaccuracies sometimes make it difficult to understand. Several times, the text and figures are inconsistent.

Problems:

Line 141: DO should be OD

Lines 172-175:

“After incubation, was added peroxidase-conjugated goat anti-human IgG (1:2000) antibody, biotinylated anti-IgG1, anti-IgG3 and anti-IgG4 (Sigma Chemical Co., St. Louis, USA) biotinylated detection antibodies diluted 1: 2000, 1:2000 and 1:4000, respectively.” – This sentence is confusing.

Figure 2

Line 227-228: “Evaluating ELISA index IgG1 levels (median = 5.35) presented statistically higher levels than IgG3 (median = 1.36) (p < 0.0001) and IgG4 (median = 1.39) (p < 0.0001) in serum. – According to Figure 2, the median value for IgG1 in serum is about 4.3.

Line 229-231: “In colostrum IgG3 (median = 2.60) presented higher levels than IgG1 (median = 2.45) (p < 0.05) and IgG4 (median = 1.81) (p < 0.001).” – These median values also seem to be different from those on Fig. 2.

Lines 232-235: “Besides that, the levels of ELISA index from IgG1 and IgG4 antibodies T. gondii specific detected in colostrum samples were significantly higher than in paired serum samples (p < 0.001). There was no statistical difference between IgG4 levels in serum and colostrum (p = 0.9146).” - These sentences are contradictory with each other and with Fig. 2.

lines 277-281: These sentences are grammatically incorrect.

lines 285-286: “In colostrum IgG1 levels were 285 statistically higher than IgG3 (p <0.05) and IgG4 (p < 0.01).” –  It seems to me that the value of IgG3 is the highest in Figure 4B.

lines 322-323: “In colostrum, however, less than half of the samples (56.9%) were positive for IgG1.”  - 56.9% is more than half.

lines 369-372: This sentence is grammatically incorrect.

Author Response

Author's Reply to the Review Report (Reviewer 1)

Comments and Suggestions for Authors

“This is an interesting article, it would be worth publishing, however, linguistic inaccuracies sometimes make it difficult to understand. Several times, the text and figures are inconsistent.”

We would like to thank the reviewer for the attention and comments throughout the manuscript, which will improve the understanding of the message to the readers.

Problems:

Line 141: DO should be OD

This correction was done.

Lines 172-175:

After incubation, was added peroxidase-conjugated goat anti-human IgG (1:2000) antibody, biotinylated anti-IgG1, anti-IgG3 and anti-IgG4 (Sigma Chemical Co., St. Louis, USA) biotinylated detection antibodies diluted 1: 2000, 1:2000 and 1:4000, respectively.” – This sentence is confusing.

This sentence was rewritten.

Figure 2

Line 227-228: “Evaluating ELISA index IgG1 levels (median = 5.35) presented statistically higher levels than IgG3 (median = 1.36) (p < 0.0001) and IgG4 (median = 1.39) (p < 0.0001) in serum. – According to Figure 2, the median value for IgG1 in serum is about 4.3.

This sentence was corrected, as follows: “ELISA index to evaluate IgG1 levels in serum samples (median = 4.23) presented statistically higher values, when compared to IgG3 and IgG4 levels (medians = 1.20 and 1.13, respectively) (p < 0.0001).”

Line 229-231: “In colostrum IgG3 (median = 2.60) presented higher levels than IgG1 (median = 2.45) (p < 0.05) and IgG4 (median = 1.81) (p < 0.001).” – These median values also seem to be different from those on Fig. 2.

In colostrum samples, however, IgG3 presented higher levels (median = 1.93) than IgG1 (median = 1.28) (p < 0.05) and IgG4 (median = 1.05) (p < 0.001). Concerning the comparison between IgG1 and IgG4 levels in colostrum a significant difference was also found (p < 0.01).

Lines 232-235: “Besides that, the levels of ELISA index from IgG1 and IgG4 antibodies T. gondii specific detected in colostrum samples were significantly higher than in paired serum samples (p < 0.001). There was no statistical difference between IgG4 levels in serum and colostrum (p = 0.9146).” - These sentences are contradictory with each other and with Fig. 2.

These sentences were rephrased, as follows: “Moreover, the levels of ELISA index for IgG1 antibodies against T. gondii detected in serum samples were significantly higher when compared to paired samples of colostrum (p < 0.0001). In contrast, significantly higher levels of ELISA index for IgG3 antibodies to this parasite were found in colostrum samples (p < 0.0001), when compared to paired serum samples. There was no statistical difference between IgG4 levels in serum and colostrum samples (p = 0.9146).”

lines 277-281: These sentences are grammatically incorrect.

These sentences were corrected.

lines 285-286: “In colostrum IgG1 levels were 285 statistically higher than IgG3 (p <0.05) and IgG4 (p < 0.01).” –  It seems to me that the value of IgG3 is the highest in Figure 4B.

The Reviewer is right. This typing mistake was corrected in the new version of the manuscript.

lines 322-323: “In colostrum, however, less than half of the samples (56.9%) were positive for IgG1.”  - 56.9% is more than half.

This sentence was corrected, as indicated.

lines 369-372: This sentence is grammatically incorrect.

This sentence was rephrased.

Reviewer 2 Report

The paper entitled “Comparative detection of immunoglobulin isotypes against Toxoplasma gondii soluble antigen in serum and colostrum samples from puerperal women” by Borges et al, describes the quantification of IgG, IgG1. IgG3, IgG4, IgM and IgA in serum and colostrum samples of 283 postpartum women from Uberlândia, Brazil, and relates results with diagnosis status.

The text is clearly written and easy to read. However, it is necessary to make some corrections in the English writing.

Major comments

Some sections of the article are not sufficiently developed. For example, the results presented in table 3 are not reflected in the summary, materials and methods and discussion. In the Minor comments, all points that need improvement will be addressed.

Minor comments

ABSTRACT

Lines 22-24: The Title of the article refers to anti-T. gondii immunoglobulin isotype detection, however the Abstract only refers to subclass detection (IgG1, IgG3, IgG4). I think it is important to mention in the subsection Methods of the Abstract the detection of immunoglobulin isotypes/classes (IgG, IgM, IgA).

Lines 24-34: The results in Tables 1 and 2 are not reflected in the abstract (nor are they discussed in the Discussion section). Are these results not relevant?

INTRODUCTION

The Introduction does not mention the kinetics of antibody production after primary infection nor the serological diagnosis of acute and chronic T. gondii infection, which may make the article difficult for some readers.

MATERIAL AND METHODS

The description of groups I, II and III must be presented in this section, as well as the diagnosis status.

Lines 88-90: Needs correction of English writing

Lines 114-121: Isolation and purification of T. gondii tachyzoite forms from cell cultures to obtain antigen is not sufficiently clear. The antigen preparation protocol was not explained. I would like the authors to clarify in the text how they obtain free parasites and remove cellular debris, and how to prepare the T. gondii antigen (STAg).

Lines 161-180: The indirect ELISA for detection of IgG anti-T. gondii was already described in Lines 123-136. The problem can be solved in two ways: remove the text referring to IgG from this subsection or describe the ELISA protocol for detection of IgG simultaneously with the protocol for detecting IgG subclasses.

RESULTS

Line 194-196: Needs correction of English writing.

Table 2: In the Materials and Methods section, it is not mentioned how the information regarding the diagnosis status was obtained.

DISCUSSION

Results presented in Tables 1 and 2 were not adequately discussed.

Lines 326-328: The sentence needs to be rewritten, because it is not understandable. What do you mean with “compensating factor”?

Lines 329-330: The sentence needs to be rewritten. “When we analyzed the profile”- Does it refer to the present study?

Lines 334-336: Needs correction of English writing.

Line 339: I suggest replacing the expression “responded to”.

Author Response

Author's Reply to the Review Report (Reviewer 2)

Comments and Suggestions for Authors

The paper entitled “Comparative detection of immunoglobulin isotypes against Toxoplasma gondii soluble antigen in serum and colostrum samples from puerperal women” by Borges et al, describes the quantification of IgG, IgG1. IgG3, IgG4, IgM and IgA in serum and colostrum samples of 283 postpartum women from Uberlândia, Brazil, and relates results with diagnosis status.

We appreciated the time and attention of the Reviewer to include the suggestions and comments throughout the manuscript.

The text is clearly written and easy to read. However, it is necessary to make some corrections in the English writing.

I agree with the Reviewer that the mentioned corrections will improve the understanding of the message to the readers.

Major comments

Some sections of the article are not sufficiently developed. For example, the results presented in table 3 are not reflected in the summary, materials and methods and discussion. In the Minor comments, all points that need improvement will be addressed.

The new version of this manuscript was entirely reviewed to emphasize the relevant pieces of information in all sections, from the abstract to the discussion sections.

Minor comments

ABSTRACT

Lines 22-24: The Title of the article refers to anti-T. gondii immunoglobulin isotype detection, however the Abstract only refers to subclass detection (IgG1, IgG3, IgG4). I think it is important to mention in the subsection Methods of the Abstract the detection of immunoglobulin isotypes/classes (IgG, IgM, IgA).

We agree with the Reviewer. The subsection Methods of the Abstract section was rewritten to specify the names of the isotypes and subclasses of the antibodies.

Lines 24-34: The results in Tables 1 and 2 are not reflected in the abstract (nor are they discussed in the Discussion section). Are these results not relevant?

As above mentioned, the new version of this manuscript was entirely reviewed to emphasize the relevant pieces of information in all sections, from the abstract to the discussion sections.

INTRODUCTION

The Introduction does not mention the kinetics of antibody production after primary infection nor the serological diagnosis of acute and chronic T. gondii infection, which may make the article difficult for some readers.

We agree with the Reviewer. The new version of the manuscript now contains these pieces of information concerning the kinetics of antibody production to characterize the acute versus chronic phases of T. gondii infection.

MATERIAL AND METHODS

The description of groups I, II and III must be presented in this section, as well as the diagnosis status.

Groups classification and status diagnosis

Classically, samples with positivity only for IgG antibodies are related to the chronic phase. The acute phase of infection is serologically characterized by the concomitant presence of IgG, IgM and IgA antibodies, or the presence of IgG and IgM, or the presence of IgM and IgA in the absence of IgG antibodies.

To better study the serological results based on the reactivity IgG1, IgG3 and IgG4 T. gondii specific, the samples were divided into three different groups: Group 1: serum and colostrum samples of puerperal women with acute T. gondii infection, characterized by positives results for specific IgG, IgM and IgA antibodies (IgG+/ IgM+/IgA+). Group 2: serum and colostrum samples of puerperal women with positive results for IgG and IgM antibodies specific for T. gondii (IgG+/ IgM+/IgA-). Group 3: serum and colostrum samples of puerperal women with chronic T. gondii infection, characterized by only positive results for specific IgG (IgG+/ IgM-/IgA-).

To better define the diagnostic status, IgG1 and IgG3 antibodies were evaluated, characterizing two possible outcomes: early acute phase and convalescent phase, as previously described. The early acute phase (0-3 months of infection) has an IgG3/IgG1 ratio > 1.0 with positive IgM and/or IgA antibodies. The convalescent phase (3-12 months of infection) has an IgG3/IgG1 ratio < 1.0 with positive IgM and/or IgA.

Lines 88-90: Needs correction of English writing

These sentences were corrected, as suggested.

Lines 114-121: Isolation and purification of T. gondii tachyzoite forms from cell cultures to obtain antigen is not sufficiently clear. The antigen preparation protocol was not explained. I would like the authors to clarify in the text how they obtain free parasites and remove cellular debris, and how to prepare the T. gondii antigen (STAg).

As requested by the Reviewer, the reference containing the complete protocol used to remove cellular debris and obtain free parasites was included in the new version of the manuscript (Ribeiro DP et al. https://doi.org/10.1016/j.vaccine.2009.02.028).

Lines 161-180: The indirect ELISA for detection of IgG anti-T. gondii was already described in Lines 123-136. The problem can be solved in two ways: remove the text referring to IgG from this subsection or describe the ELISA protocol for detection of IgG simultaneously with the protocol for detecting IgG subclasses.

As suggested, the protocols for the detection of total IgG and subclasses by indirect ELISA are now described simultaneously in the new version of the manuscript.

RESULTS

Line 194-196: Needs correction of English writing.

The mentioned phrases were corrected.

Table 2: In the Materials and Methods section, it is not mentioned how the information regarding the diagnosis status was obtained.

 In this new version of the manuscript it was added a new section entitled “Groups classification and status diagnosis” in order to clarify how the information concerning the diagnosis status from all groups of samples was defined.

DISCUSSION

Results presented in Tables 1 and 2 were not adequately discussed.

In the new version of the manuscript, it was inserted a new table to clarify the points necessary to be emphasized and discussed adequately.

Lines 326-328: The sentence needs to be rewritten, because it is not understandable. What do you mean with “compensating factor”?

This sentence was rewritten to: “It can be hypothesized that, due to this fact, there is a lower presence of antibodies T. gondii specific from this immunoglobulin subclass in maternal colostrum, being considered as a counterbalancing factor, since there is a high maternal-neonatal transfer of IgG1 through the placenta.”

Lines 329-330: The sentence needs to be rewritten. “When we analyzed the profile”- Does it refer to the present study?

In fact, the information refers to the results from the reference [43]. To make clear this information to the readers, this sentence was rewritten as follows: When analyzing the detection of IgG subclasses in colostrum in comparison with the total IgG content, it has already been described that IgG1 was predominant, followed by IgG2, IgG4 and IgG3 [43].”

Lines 334-336: Needs correction of English writing.

This sentence was rephrased to: “Similarly, the data from the present report reinforce the hypothesis that the presence of detectable levels of IgG3 in colostrum may be the result of  a selective secretion and/or due to a local production of this subclass by the mammary gland cells.”

Line 339: I suggest replacing the expression “responded to”.

As suggested, the expression was replaced for: “…..detect with a higher percentage of accuracy the cases of a mother infected by T. gondii based also in the seropositivity for IgG3.

Round 2

Reviewer 2 Report

The paper entitled “Comparative detection of immunoglobulin isotypes against Toxoplasma gondii soluble antigen in serum and colostrum samples from puerperal women” by Borges et al, describes the quantification of IgG, IgG1. IgG3, IgG4, IgM and IgA in serum and colostrum samples of 283 postpartum women from Uberlândia, Brazil, and relates results with diagnosis status.

I remind the authors of the need to correct the English language. The text would greatly benefit if corrected by an English speaker.

Although the corrections that have been suggested have been introduced in the text, I point out some details that need correction.

I also ask the authors to make a good review of the entire text.

Minor comments

ABSTRACT

Lines 23-25: I suggest- “ELISA immunoassays for detection of anti-T. gondii-specific IgM, IgA, IgG isotypes and IgG1, IgG3 and IgG4 subclasses were conducted on paired samples of serum and colostrum.”

Lines 28: The sentence needs to be rewritten. I suggest: “Thus, the predominant reactivity of IgG subclasses against T. gondii was….”

Lines 36-39: I suggest removing the word “subclasses”.

INTRODUCTION

Lines 53-55: I suggest putting this sentence in the previous paragraph and starting a new paragraph with: "During..."

MATERIAL AND METHODS

Lines 96-98: Sentence needs to be rewritten. I suggest: “The present study involved 283 postpartum women, who were hospitalized due to childbirth at The Obstetric Center of the Hospital de Clínicas da Universidade Federal de Uberlândia and accepted to participate in the study after being duly clarified.”

Lines 96-98: I suggest that the text "Groups classification and status diagnosis” be moved to the first paragraph of “Statistical Analysis”. It may not be necessary to keep the first sentence of this bit of text, but I leave the decision to the authors.

RESULTS

Line 269: “…anti-T. gondii in serum and colostrum….”.

DISCUSSION

Lines 350-353: I still don't understand the explanation.

Line 363: “…(54.6%) were reactive…”

Lines 384-390: The paragraph contains important information, which was not present in the previous version, but needs to be rewritten to improve the English language.

Author Response

Answers to Reviewer 2:

The paper entitled “Comparative detection of immunoglobulin isotypes against Toxoplasma gondii soluble antigen in serum and colostrum samples from puerperal women” by Borges et al, describes the quantification of IgG, IgG1. IgG3, IgG4, IgM and IgA in serum and colostrum samples of 283 postpartum women from Uberlândia, Brazil, and relates results with diagnosis status.

We thank again the Reviewer for the careful revision of this manuscript, which will clarify the message of our work to the readers.

I remind the authors of the need to correct the English language. The text would greatly benefit if corrected by an English speaker.

We agree with this suggestion and the new version of the manuscript was thoroughly reviewed by an English speaker.

Although the corrections that have been suggested have been introduced in the text, I point out some details that need correction.

These corrections were done.

I also ask the authors to make a good review of the entire text.

 The entire text was reviewed, as requested.

Minor comments

ABSTRACT

Lines 23-25: I suggest- “ELISA immunoassays for detection of anti-T. gondii-specific IgM, IgA, IgG isotypes and IgG1, IgG3 and IgG4 subclasses were conducted on paired samples of serum and colostrum.

This statement was rephrased, as suggested.

Lines 28: The sentence needs to be rewritten. I suggest: “Thus, the predominant reactivity of IgG subclasses against T. gondii was….

This statement was rephrased, as suggested.

Lines 36-39: I suggest removing the word “subclasses”.”

The word “subclasses” was removed, as suggested.

INTRODUCTION

Lines 53-55: I suggest putting this sentence in the previous paragraph and starting a new paragraph with: "During...

These modifications were carried out in both paragraphs, as stated.

MATERIAL AND METHODS

Lines 96-98: Sentence needs to be rewritten. I suggest: “The present study involved 283 postpartum women, who were hospitalized due to childbirth at The Obstetric Center of the Hospital de Clínicas da Universidade Federal de Uberlândia and accepted to participate in the study after being duly clarified.

This sentence was rewritten, as suggested.

Lines 96-98: I suggest that the text "Groups classification and status diagnosis” be moved to the first paragraph of “Statistical Analysis”. It may not be necessary to keep the first sentence of this bit of text, but I leave the decision to the authors.

These suggestions were all accepted and included in the new version of the manuscript.

RESULTS

Line 269: “…anti-T. gondii in serum and colostrum….”.

 This correction was done.

DISCUSSION

Lines 350-353: I still don't understand the explanation.

The sentences were rewritten, as follows: “Even though previous studies have already shown that all four IgG subclasses can cross the placenta to protect the fetus from infections, there is a higher rate of IgG1 vertical transport when compared to other IgG subclasses. In colostrum, however, only around half of the samples (56.9%) were positive for IgG1. It can be hypothesized that the lower presence of this immunoglobulin subclass in maternal colostrum antibodies specific to T. gondii, could be counterweighed by the high maternal-neonatal transfer of IgG1 anti-T. gondii through the placenta.

 “Line 363: “…(54.6%) were reactive…

This correction was done.

Lines 384-390: The paragraph contains important information, which was not present in the previous version, but needs to be rewritten to improve the English language.

 This paragraph was rewritten, as follows: “the present study included the results of 16 cases of puerperal women with diagnostic status of acute toxoplasmosis. paired samples of serum or colostrum from these patients were analyzed to characterize the diagnostic status of T. gondii infection. When assessing these results, it is clear that new pieces of information were obtained. In fact, the diagnostic status based on the results from colostrum samples of 9 cases indicated a profile of more recent infection, when compared to those obtained from serum samples; 5 showed a similar diagnostic status for both colostrum and serum samples; and 2 had a profile indicating a later diagnostic status of T. gondii infection when colostrum results where compared to serum. Taken together, these results support the idea of using colostrum samples as a useful diagnostic tool to determine the status of T. gondii infection, especially during the acute phase.”